# Crystal structure of an invertebrate cytolysin pore reveals unique properties and mechanism of assembly

Marjetka Podobnik[1], Peter Savory[2], Nejc Rojko[1], Matic Kisovec[1], Neil Wood[2], Richard Hambley[2], Jonathan Pugh[2], E. Jayne Wallace[2], Luke McNeill[2], Mark Bruce[2], Idlir Liko[3], Timothy M. Allison[3], Shahid Mehmood[3], Neval Yilmaz[4], Toshihide Kobayashi[4], Robert J.C. Gilbert[5], Carol V. Robinson[3], Lakmal Jayasinghe[2] & Gregor Anderluh[1]

The invertebrate cytolysin lysenin is a member of the aerolysin family of pore-forming toxins that includes many representatives from pathogenic bacteria. Here we report the crystal structure of the lysenin pore and provide insights into its assembly mechanism. The lysenin pore is assembled from nine monomers via dramatic reorganization of almost half of the monomeric subunit structure leading to a β-barrel pore ∼10 nm long and 1.6–2.5 nm wide. The lysenin pore is devoid of additional luminal compartments as commonly found in other toxin pores. Mutagenic analysis and atomic force microscopy imaging, together with these structural insights, suggest a mechanism for pore assembly for lysenin. These insights are relevant to the understanding of pore formation by other aerolysin-like pore-forming toxins, which often represent crucial virulence factors in bacteria.

[1] Department for Molecular Biology and Nanobiotechnology, National Institute of Chemistry, Hajdrihova 19, 1000 Ljubljana, Slovenia. [2] Oxford Nanopore Technologies Ltd., Edmund Cartwright House, 4 Robert Robinson Avenue, Oxford Science Park, Oxford OX4 4GA, UK. [3] Department of Chemistry, University of Oxford, South Parks Road, Oxford OX1 3QZ, UK. [4] Lipid Biology Laboratory, RIKEN Institute, Wako 351-0198, Japan. [5] Division of Structural Biology, Wellcome Trust Centre for Human Genetics, University of Oxford, Roosevelt Drive, Oxford OX3 7BN, UK. Correspondence and requests for materials should be addressed to L.J. (email: lakmal.jayasinghe@nanoporetech.com) or to G.A. (email: gregor.anderluh@ki.si).

Pore-forming proteins are widespread in nature and have important physiological roles in attack and defence mechanisms[1,2]. They are extremely powerful molecules destined to form pores in lipid membranes of target cells to cause killing or other undesired effects. The most studied pore-forming proteins are various families of bacterial toxins, termed pore-forming toxins (PFTs). PFTs in bacteria serve as important virulence agents promoting bacterial spread through invading cells and tissues. However, PFTs are widespread in nature and may be found also in other organisms[3,4]. Generally, they are secreted as water-soluble monomers, which upon binding to target lipid membranes, oligomerize and form transmembrane pores detrimental to cells[1]. They can be classified as either α- or β-PFTs, based on the secondary structure elements that form the transmembrane region of the pore[4]. The three most important β-PFTs families are cholesterol-dependent cytolysins found predominately in Gram-positive bacteria[5], the α-haemolysin family found predominately in *Staphylococcus aureus*[6] and PFTs similar to aerolysin from *Aeromonas hydrophila*[7]. Apart from having important roles in bacterial pathogenesis, biological nanopores have recently attracted a lot of attention as prime candidates for various applications in medicine and nanobiotechnology[8,9].

Aerolysin was the first β-PFT for which the X-ray crystal structure of the soluble form was determined, more than 20 years ago[10]. It is also the founding member of the aerolysin β-PFTs (aβ-PFTs) family, which now includes many different examples from bacteria to vertebrates (Supplementary Fig. 1) (ref. 7). Many examples of aβ-PFTs are present in pathogenic bacteria such as *Aeromonas*, *Clostridium* or *Bacillus*, where they serve as crucial virulence factors in infection and food poisoning. Members of the aerolysin family have been found to form hepta-, octa- or nonameric oligomers, as reported for aerolysin, Dln1 (zebrafish aβ-PFTs) and monalysin (aβ-PFTs from *Pseudomonas entomophila*), respectively[11–13]. However, they share similar overall dimensions of the pore complex and the lumen of the pore. The molecular details of pore assembly by aβ-PFTs have not been known in atomic detail, up to now.

One of the few described eukaryotic members of the aβ-PFT family is lysenin, an invertebrate member present in the earthworm *Eisenia fetida*. Lysenin is present in the coelomic fluid of earthworms and is produced to act defensively against parasitic microorganisms by forming pores in sphingomyelin (SM)-containing membranes[14,15]. Lysenin has a rare capacity to bind plasma membrane lipid SM with high affinity[16]. Together with other natural toxins, such as equinatoxin II from sea anemone, it represents an excellent tool for visualizing distribution and dynamics of SM in cells[17–20]. Monomeric lysenin consists of two distinct domains, the elongated N-terminal domain (pore-forming module, PFM) that shares the fold of all aβ-PFTs (Supplementary Fig. 1), and the C-terminal β-trefoil lectin type domain[21]. Here we report the X-ray crystal structure of the lysenin pore. We reveal that lysenin forms pores using nine subunits assembled to form to a large extent a smooth-walled tubular β-barrel structure flanked on the outside by the C-terminal domains. We also provide insights into assembly of the lysenin pore relevant to other aβ-PFTs.

## Results

**The lysenin pore is a nonamer.** When exposed to SM-containing membranes, lysenin forms oligomers of high stability, resistant to heating in excess of 95 °C and high SDS concentrations[22]. In addition, once purified from SM-containing membranes, lysenin pores are readily inserted into the artificial membrane of the MinION platform developed for DNA sequencing (Supplementary Fig. 2). We attempted to develop lysenin as a DNA sensor, however, DNA failed to translocate through the wild-type lysenin oligomer (Supplementary Fig. 2). Based on the crystal structure of the monomeric lysenin, five negative charges were removed from the proposed transmembrane region (Glu84Gln, Glu85Lys, Glu92Gln, Glu97Ser and Asp126Gly) to create the mutant RL1. RL1 still readily forms stable oligomers in the presence of SM-containing membrane and successfully captures and translocates DNA strands under an applied voltage potential (Supplementary Fig. 2).

The homogeneous RL1 pore sample was obtained by exposing it to SM-containing membranes. For structural studies, RL1 oligomers were solubilized from liposomes containing SM and cholesterol with *n*-dodecyl-β-D-maltoside (DDM). We explored the heterogeneity of the protein complex using mass spectrometry to determine the extent of oligomerization, the presence of lipids and the stability in different detergents[23]. A mass spectrum of the intact RL1 oligomer from a DDM micelle (Fig. 1a) revealed species with a mass matching that of nine times the predicted monomeric mass, indicating the RL1 oligomer is a nonamer. Further, collision-induced dissociation products, a monomer and octamer, are also present in the spectrum, and derive from

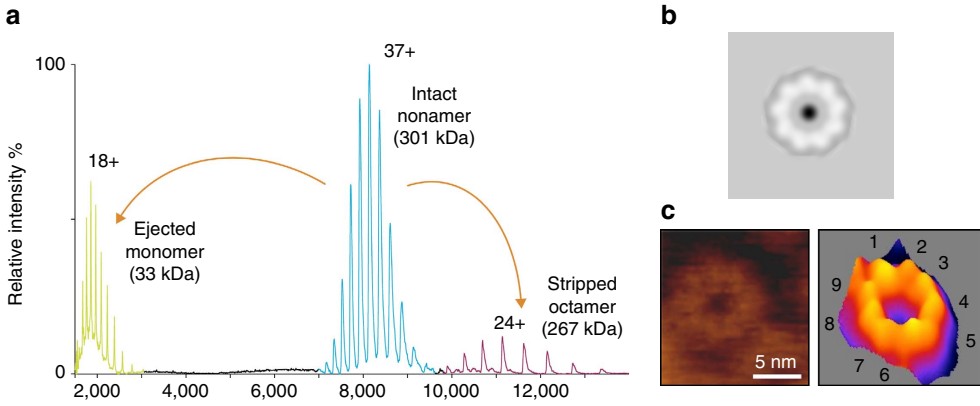

**Figure 1 | The lysenin pore is a nonamer.** (**a**) Mass spectrum of intact RL1 pore. Activation of nonameric species in the gas phase by collision-induced dissociation resulted in a monomer and octamer providing additional evidence for the presence of the nonamer in solution. (**b**) The wild-type lysenin pore, as previously reported from two-dimensional electron crystallographic analysis[21], with ninefold symmetry. (**c**) AFM image of wild-type lysenin oligomer on sphingomyelin/cholesterol (1:1) bilayer with its three-dimensional image to the right. The numbers indicate the lysenin protomers.

the nonameric species. Independent of the measured mass of the nonameric species matching, these products confirm its stoichiometry. Notably, absent in the mass spectrum of lysenin is the presence of any lipid adducts, commonly observed in spectra of other membrane proteins, which coupled with the single oligomeric state, suggests a high degree of homogeneity suitable for structural and functional studies. The nonameric architecture of RL1 in DDM micelles is biologically relevant, as independent imaging of the wild-type lysenin oligomers formed on the surface of lipid membranes rich in SM by two-dimensional electron crystallographic analysis (Fig. 1b) or atomic force microscopy (AFM; Fig. 1c) also showed nonameric assemblies. These results underscore the diversity in aβ-PFT pore stoichiometry. Aerolysin and the zebrafish aerolysin-like protein form pores using seven[11] and eight[12] monomers, respectively. However, as in the case of lysenin, a nonameric pore structure was also suggested for monalysin[13].

**Lysenin pore is exclusively a β-barrel.** The homogeneous RL1 nonamer was crystalized and its structure determined at 3.1 Å resolution (Table 1). Two nonameric oligomers were present in the asymmetric unit, packed in a head-to-head manner (Supplementary Fig. 3). The RL1 nonamer reveals a mushroom-shaped transmembrane pore with a central stem built of a long β-barrel (Fig. 2, Supplementary Movie 1). Every protomeric subunit of the two pores is completely defined, except for 9 N-terminal residues, and the loops connecting the β-hairpins at the *trans* side of the barrel that could be traced in only two protomers in one (chains A-I) and six in the other (chains J-S) pore. The bottom half of the mushroom cap is composed of the C-terminal domains of each protomer, and the upper part, named the collar, of β-strands originating from the N-terminal domains. There is no interaction between C-terminal domains and the

β-barrel (Fig. 2). The β-barrel is located in the middle of the oligomeric assembly and is built by nine β-hairpins, each resulting from the unfurling of a significant portion of the N-terminal domain of each protomer. The barrel extends from the top to the bottom of the pore, which is a feature unique to this structure among those available for β-PFT pores (Fig. 2). The height of the pore is ~11 nm, of which 10 nm belongs to the β-barrel, with the widest dimension of the cap 12.3 nm. The inner diameter of the pore ranges from 1.6 to 2.5 nm (Fig. 3a). The inner surface of the RL1 channel is mostly polar, with negatively charged residues framing the top and the bottom (Fig. 3b). If mutated residues in RL1 are modelled back to their wild-type counterparts, the inner surface of the channel becomes almost completely negatively charged (Supplementary Fig. 4), thus neatly explaining the reported exclusivity for the passage of cations[24] as well as the inability of the wild-type lysenin pores to translocate DNA (Supplementary Fig. 2).

One of the aβ-PFTs prominent structural features are tracts of Ser and Thr residues that were suggested to participate in membrane binding[25], promote oligomerization[26] or guide the amphipathic loops towards the membrane during pore formation[11]. The crystal structure now reveals that Ser and Thr residues are distributed along β-hairpins of the entire tube, not only in the transmembrane part (Fig. 3c). Most of their side chains face the lumen, except for some patches on the outside of the barrel in the extramembrane region, where they point into the space between the barrel and the cap. With their polar and relatively short side chains they may contribute to the hydrogen bond network and thus stability of the barrel structure, as well as its overall polarity and shape.

The shape of the pore as well as the distribution of the charged and hydrophobic residues indicates its positioning and orientation in membranes. The relatively flat surface of the bottom part of the cap contains positively and negatively charged cavities that could fit the charged phosphocholine (POC) heads of SM. Superposition of the soluble monomer (PDB-ID 3ZX7) with bound POC molecules reveals how two POC groups fit into the cavities in the membrane-interacting part of the cap with a third one to the side of the cap, also relatively close to the membrane surface (Fig. 3d). Such multivalent binding of membrane lipid receptors is not common in PFTs and explains nicely lysenin's affinity for clustered SM and the capacity to form oligomers in SM-rich domains[20,27].

Residues Val34-Ile107 of each protomer form a long and tilted β-hairpin. The part of the β-hairpin loop that protrudes below the flat bottom of the cap includes residues Val54 to Ser86. This part of the barrel is ~4 nm high (Fig. 2) and rich in

| | Lysenin pore |
|---|---|
| **Table 1 | Data collection and refinement statistics.** | |
| *Data collection* | |
| Space group | P4$_3$2$_1$2 |
| Cell dimensions | |
| *a, b, c* (Å) | 193.0, 193.0, 493.2 |
| *α, β, γ* (°) | 90.0, 90.0, 90.0 |
| Resolution (Å) | 50.00–3.10 (3.29–3.10)* |
| $R_{sym}$ | 20.4 (80.4) |
| $I/\sigma I$ | 8.9 (2.3) |
| CC$_{1/2}$ (%) | 99.6 (87.3) |
| Completeness (%) | 99.9 (99.7) |
| Redundancy | 8.1 (8.3) |
| | |
| *Refinement* | |
| Resolution (Å) | 49.1–3.1 |
| No. of reflections | 168,954 |
| $R_{work}/R_{free}$ (%) | 21.44/25.79 |
| No. of atoms | |
| Protein | 40,953 |
| Ligand/Ions | 234 |
| Water | 61 |
| B-factors | |
| Protein | 57.3 |
| Ligand/Ions | 81.1 |
| Water | 27.8 |
| Root mean square deviations | |
| Bond lengths (Å) | 0.009 |
| Bond angles (°) | 1.475 |

*Values in parentheses are for the highest-resolution shell. One crystal was used to measure the data.

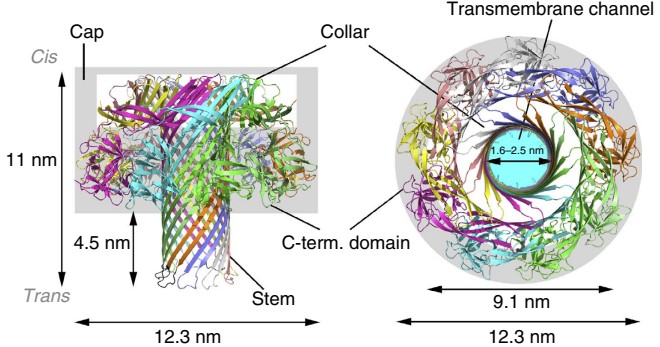

**Figure 2 | The lysenin pore structure.** Overall ribbon representation of the lysenin pore coloured by chain. Missing loops at the bottom of the β-barrel were modelled based on the determined ones and are shown as black ribbons. Left: side view, right: top view.

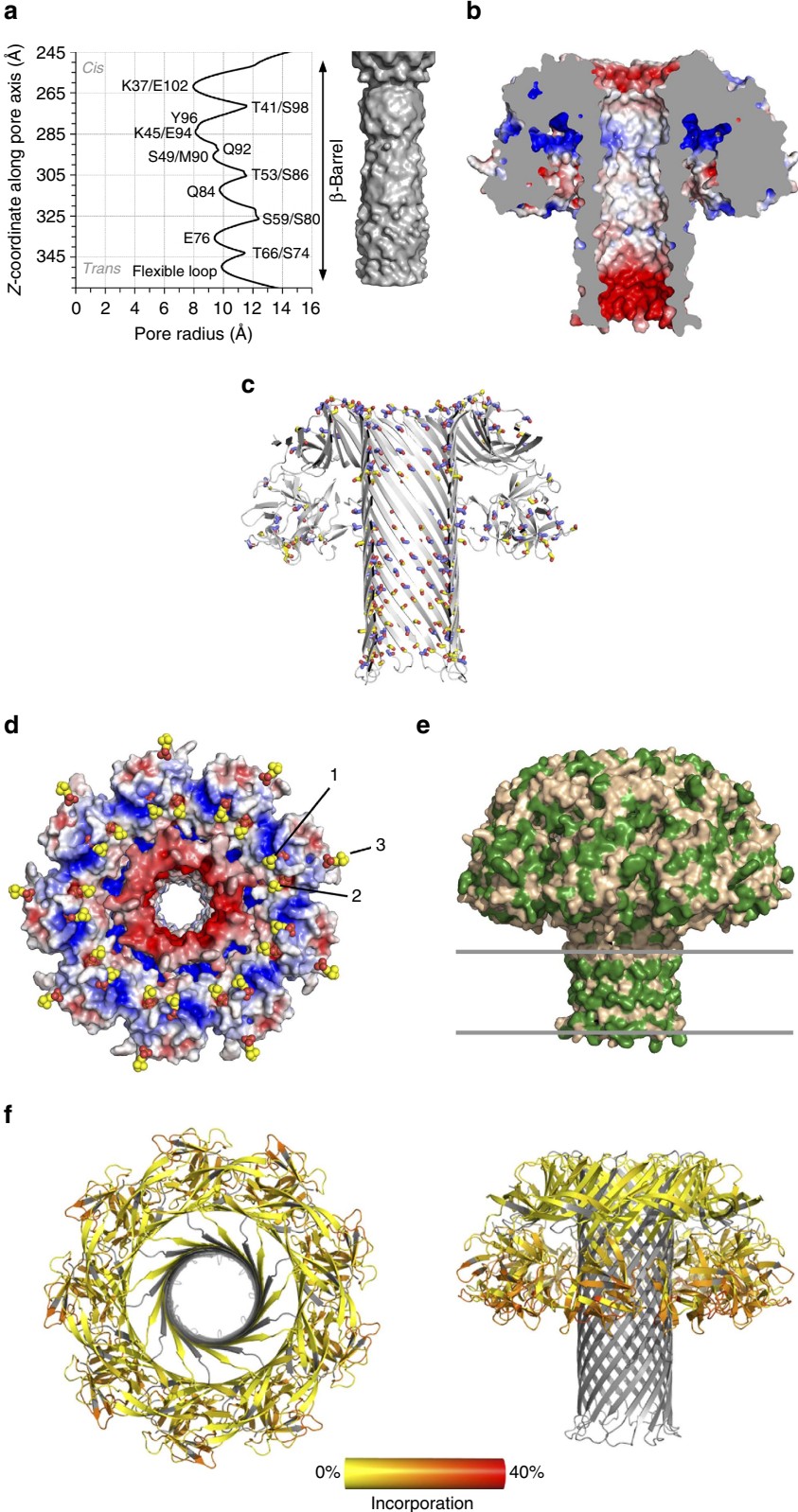

**Figure 3 | Properties of the lysenin pore.** (**a**) Channel radius profile and internal shape of the channel. (**b**) Electrostatic surface of the RL1 pore, side view. Cutoff $-5\,kTe^{-1}$ was used for negative potential (red) and $+5\,kTe^{-1}$ for positive potential (blue). (**c**) The distribution of Thr (blue-red sticks) and Ser (yellow-red sticks) residues in the pore structure. (**d**) The same as in **b** viewing the cap part facing the membrane. Phosphocholine molecules (labelled with numbers 1–3) are superimposed from the soluble lysenin structure PDB-ID 3ZX7 and are depicted as spheres. (**e**) Hydrophobic (green) residues on the lysenin pore surface. Position of the membrane bilayer is depicted by lines. (**f**) Relative fractional uptake measured by HDX-MS mapped to the structure of lysenin pore. Regions coloured grey have no sequence coverage for the peptide mapping. Left: top view, right: side view.

hydrophobic residues (Fig. 3e). We independently assessed RL1 pore membrane interactions by hydrogen/deuterium exchange mass spectrometry (HDX)[28]. HDX was used to examine the shape and position of the RL1 pore extracted from the liposomes in the DDM micelle. HDX clearly showed that there was no sequence coverage and/or slow exchange in the proposed β-barrel region, probably due to hindrance by the DDM micelle. By contrast, the C-terminal domains in the cap underwent exchange most readily, consistent with their high conformational dynamics in solution and exposure to the solvent (Fig. 3f).

**Lysenin undergoes structural changes during pore formation**. The crystal structure of the pore now offers an opportunity to assess the mechanism of lysenin pore assembly with broad implications for other members of the αβ-PFT family. Comparison of the structure of the monomeric soluble lysenin and protomers of the pore reveals dramatic conformational changes in the structure of lysenin upon pore formation. The structure of each pore protomer breaks down into three parts (Fig. 4a). The N-terminal domain seen in the monomeric lysenin undergoes a conformational rearrangement upon pore formation, extending strands β3 and β7 by reorganization of the segment including three β-strands (β4, β5, β6) and a single $3_{10}$ α-helix (altogether representing 25% of residues in the lysenin monomer) to reassemble into the twisted β-hairpin of the β-barrel. To form the β-hairpin between strands β3 and β7, this segment has to unfold completely in each protomer and then reassemble to form the stack of tilted and twisted β-hairpins in the final β-barrel. The other half of the N-terminal domain, a twisted β-sheet including strands β2, β8, β10 and β11, remains structurally relatively intact in comparison to the monomeric lysenin (Fig. 4a,b). However, the angle between β11 and the C-terminal domain strand changes dramatically upon pore formation, tilting β2, β8, β10 and β11 as a unit by ∼45° (Fig. 4a). Consequently, these twisted β-sheets from nine protomers now build a collar that connects the transmembrane part to the C-terminal domains in the cap of the pore (Fig. 2). The fold of the C-terminal domain is completely preserved in the pore in comparison to the monomeric lysenin (root mean square deviations of 0.341 Å between Cα-atoms of the region Gln160-Gly297; Fig. 4c). Lysenin specifically binds to SM in lipid bilayers by using the C-terminal domain[16,21]. Each C-terminal domain shares ∼2 × 500 Å$^2$ (∼15%) of its overall surface as an interface with C-terminal domains of neighbouring protomers. This clearly indicates the dual role of the C-terminal domain: it serves as an initial site of protein contact with the lipid membrane and is also important for oligomerization.

**Lysenin forms a prepore**. AFM studies previously revealed two distinct types of lysenin oligomeric structures formed on the surface of the lipid membrane[29]. These forms differ by ∼2.5 nm in height, but are similar in diameter, which is around 12 nm (11.5 ± 0.9 nm (*n* = 70) and 11.8 ± 0.9 nm (*n* = 30) for the short and tall forms, respectively; Fig. 5a,b). The taller of the two could correspond to the prepore oligomeric state, as the height corresponds to the length of the monomeric lysenin, implying that lysenin is oriented in a straight position with its C-terminal domain contacting the membrane and its PFM oriented away from the membrane[30]. If the monomeric lysenin is superimposed on C-terminal domains of the pore structure, the resulting oligomeric structure may indeed correspond to a prepore as seen by AFM (Fig. 5c). There are no clashes between lysenin protomers in this model, and the height of the model is ∼2.5 nm taller than the pore, whereas the outer diameter stays the same, both in agreement with the AFM data (Fig. 5a,b).

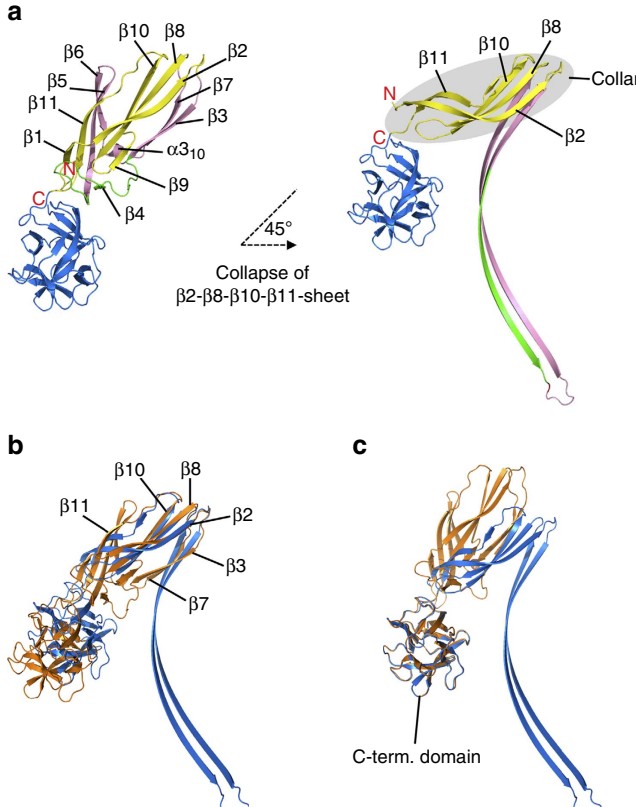

**Figure 4 | Conformational rearrangements of lysenin monomer.**
(**a**) Ribbon representation of soluble monomeric lysenin (PDB-ID 3ZXG; left). N-terminal domain: yellow (structurally intact region), pink (parts undergoing large conformational changes), green: tongue (Met44-Gly67). C-terminal domain: blue. Also shown (right) is the protomer structure from the lysenin pore, using the same colours to mark the corresponding residue ranges in the pore. (**b**) Superposition of β-sandwich of PFM and (**c**) the C-terminal domains of the soluble lysenin (PDB-ID 3ZXG, orange) on the pore protomer (blue). Superposition of Cα atoms of residues Asp15-Ile40 (**b**) and superposition on residues Phe200-Ser250 (**c**).

We propose that pore formation starts with the binding of the lysenin monomer with its C-terminal domain to an SM-rich patch in a lipid bilayer. Oligomerization proceeds with addition of other monomers to form arc-like oligomers on the surface of the membrane, with the same curvature as the prepore and as imaged by AFM (arc diameter 11.3 ± 0.8 nm (*n* = 30); Fig. 5d), which eventually grow into complete prepore rings. The conversion from a prepore to pore is possible only with significant structural rearrangement within the N-terminal domains of the protomers. In the prepore model, residues Val34-Ile107 forming the β-barrel, face the prepore lumen, thus granting enough space for the PFM to unfurl. One of the first steps after oligomerization must be the displacement of the insertion loop, the tongue (residues Met44-Gly67; Fig. 4a and Supplementary Movie 2) from the rest of the molecule, which would in turn allow breakage of H-bonds between strands of the insertion loop and the linker region (pairs β3-β9 and β5-β11). The important difference between the prepore and pore states is in the position of the N-terminal twisted β-sheet, which remains structurally intact compared with the soluble state of lysenin (Fig. 4a,b), and needs to tilt by 45° to obtain its final position in the pore state. Furthermore, the lysenin pore structure also reveals that the beginnings of the β-hairpins are already properly aligned when all the subunits come together. This provides a nucleus for β-barrel formation and the confined

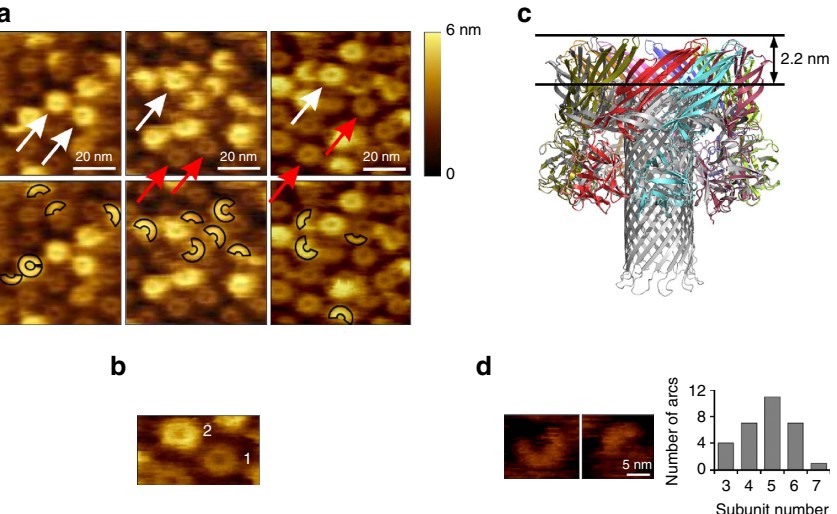

**Figure 5 | AFM analysis of lysenin prepores, pores and arcs.** (**a**) Top row: AFM height images of lysenin oligomers in prepore (white arrows) and pore states (red arrows). Bottom row: The incomplete oligomers in prepore state are marked by black-line arcs on the same images. (**b**) A representative AFM image of prepore (2) and pore (1). (**c**) Model of the lysenin prepore. The soluble lysenin structure (PDB-ID 3ZXD) was superimposed onto C-terminal domains of all protomers of the pore. The prepore model is coloured by chain and the pore is in grey. (**d**) AFM height images of arcs and the graph showing the subunit number for the arcs analysed. The arc diameter is close to that of the complete oligomers. The analysed arcs mostly consist of five subunits.

space within the oligomer would then allow efficient aligning of the remaining parts of the β-strands involved in subsequent penetration of the membrane (Supplementary Movie 2).

**Tongue region flexibility is required for pore formation**. The proposed assembly mechanism thus heavily relies on flexibility of the tongue region. The tongue is held in place by hydrophobic interactions of the side chain of Phe70, which is surrounded by several hydrophobic amino acid residues (Fig. 6a). We assessed the relative mobility of this region by molecular dynamics, showing that this is indeed one of the most flexible parts of the molecule (Fig. 6b and Supplementary Fig. 5). In an independent assay, we performed an HDX experiment and confirmed that residues Thr50 to Phe70 of the lysenin monomer readily undergo exchange (Supplementary Figs 6 and 7). The trigger for the tongue displacement from the linker region could be due to the contacts between the adjacent molecules in the prepore state and favourable interactions between the β-strands in the pore structure, especially in the collar and stem region. The β2-strand in lysenin monomers tends to bind elongated β-strand-like structures, like SM molecules[21]. In the pore, the β2-strand runs parallel to the β10-strand of the adjacent protomer in the collar. In our prepore model, the β2-strand faces the insertion loop of the neighbouring protomer. Therefore, unwinding of the tongue brings together the β2- and β10-strands of the neighbouring protomers that build upon tilting by 45° from the collar. This represents energetically and geometrically a more favourable interaction in comparison to the prepore state, and may be the trigger for prepore to pore transition.

To gain detailed information on lysenin's assembly mechanism, we made a double cysteine mutant, in which the β6-strand can be locked to the core of the monomer (β10) by a disulphide bond between Val88Cys and Tyr131Cys. Although the oxidized monomer of the double cysteine mutant was able to make oligomers, its pore formation ability was completely abolished, indicating that the oxidized monomer is locked in a prepore state (Fig. 6c). Pore-forming ability can be restored to the normal level by reducing the disulphide bond by addition of reductant to oligomers preformed on the surface of the red blood cells. These

results, therefore, clearly demonstrate that rearrangement of a significant portion of the protein chain that constitutes the final β-barrel occurs after oligomerization and prepore formation.

## Discussion

The lysenin pore structure provides a detailed overview of oligomerization-induced structural changes of αβ-PFTs. Although there seem to be subtle differences in the details of pore formation between lysenin and aerolysin, their shared mechanism is unique among PFTs. Dislocation of the tongue and the associated parts of the PFM after prepore formation is crucial for β-barrel formation in lysenin, which proceeds through tilting movement of the rigid β-sheet that, when assembled into the collar, probably serves as the starting point for barrel folding. The conformational changes thus concern almost half of the lysenin monomer and involve complete refolding of one-third of the molecule with subsequent refolding of a functional β-barrel. This is different from the archetypal β-PFT, α-haemolysin from *Staphylococcus aureus*, where the transmembrane hairpin is formed by the rearrangement of a smaller part of the protein, a pre-stem region, without any major structural disturbances in the majority of the monomeric molecule[6].

Many αβ-PFTs are synthesized with pro-peptides either on the N- or C-terminus[31]. In the case of aerolysin, this C-terminal peptide plays a dual role, namely as a chaperone for the correct folding of the soluble form, whereas its proteolysis allows oligomerization into the heptameric pore[32]. On the other hand, it was shown for monalysin that the cleavage of the N-terminal pro-peptide was required for the activation of the protein[33], enabling formation of the compact doughnut-shaped structure composed of two 9-mer prepores, which was suggested to concentrate protein in solution in the absence of the receptor-binding domain before effectively binding to the target membrane[13]. The cleavage of the pro-peptide was suggested to lead to structural rearrangements from the doughnut shape into the membrane spanning β-barrel mushroom-like nonameric pore[13]. On the other hand, no pro-peptide region is present in zebra fish Dln1 αβ-PFT. For this protein, it was suggested that in solution it exists as a soluble antiparallel dimer, which may

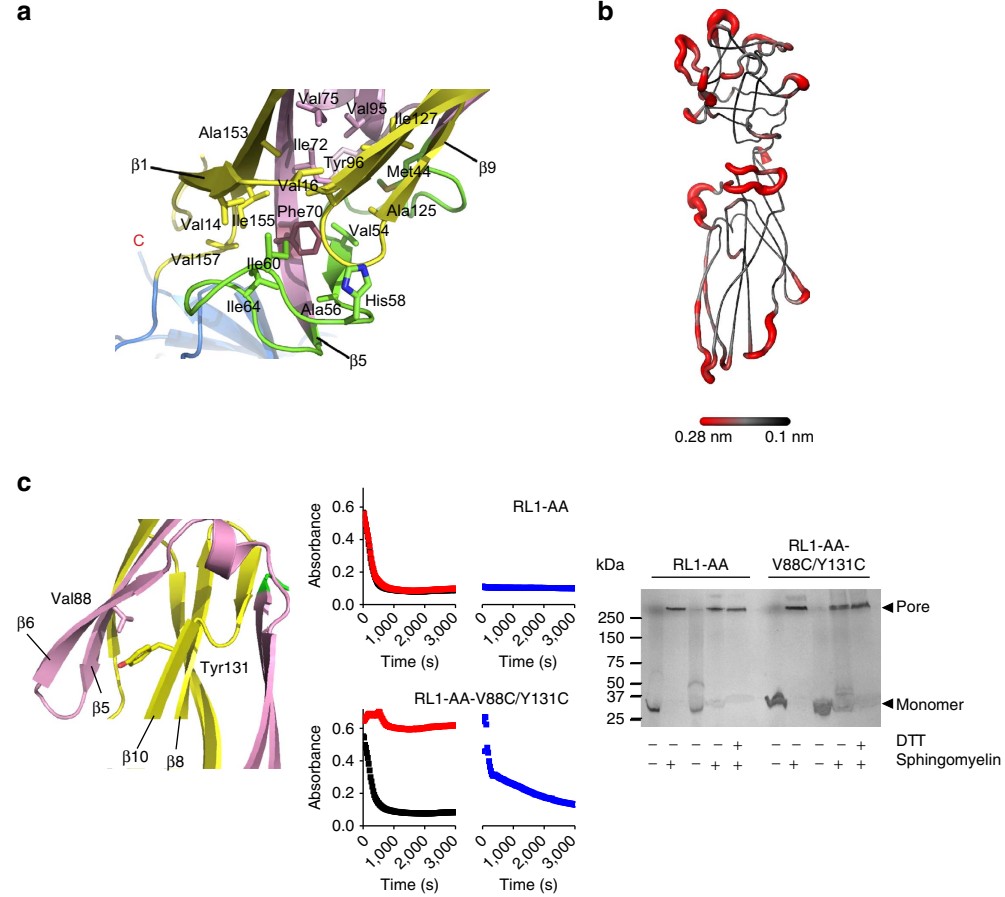

**Figure 6 | Flexibility of the tongue region and adjacent parts is crucial for pore formation. (a)** Position of Phe70 in the hydrophobic region of the soluble lysenin (PDB-ID 3ZXG). Colour code is same as in Fig. 4a. Side chains of the relevant residues are shown as sticks. C-terminus is marked as well as the relevant β-strands. (**b**) A three-dimensional model of lysenin (PDB-ID 3ZXD) with the most flexible parts of the protein backbone according to molecular dynamics simulations coloured in red. (**c**) Blocking the movement of β5 and β6 strands by forming a disulphide bond between Val88Cys-Tyr131Cys prevents pore formation. Left: representation of the region of mutation in the soluble lysenin, colour code as in **a**. Middle: haemolysis of the Lysenin-AA (two wild-type cysteines are replaced with Ala) and Lysenin-AA-V88C/Y131C. Monomers were reduced with DTT (black) or oxidized with Cu(II):1–10 Phenanthroline (red) before adding sheep red blood cells to monitor haemolysis at 595 nm. Oxidized samples were then treated with DTT and the assay repeated (blue). Right: SDS–PAGE analysis of stable oligomer formation in the presence of sphingomyelin in reduced (first two lanes for each protein sample) and oxidized (last three lanes for each protein sample) conditions.

protect the protein from oligomerization before binding to receptor. The dimer was suggested to dissociate upon membrane binding and consequently reorganize into the active pore[12]. Lysenin provides yet another possible mechanism in αβ-PFTs, as it does not contain a pro-peptide and it also does not dimerize in solution but it is monomeric. The potential explanation could be its multivalent binding site on the C-terminal domain for lipid receptors (Fig. 3d), which is unique in PFTs. This would control lysenin binding and clustering to generate an oligomeric prepore and consequently the transmembrane pore only via interaction with clustered SM in SM-rich domains[20,27].

A long β-barrel pore and the absence of any associated vestibular regions described here for the lysenin pore may also be the defining feature of other αβ-PFT pores. Lysenin is exceptional among PFTs in having a stable and uniform β-barrel that runs through the entire length of the protein[1,2]. Its high stability, unique shape and crystal structure now open ways for studies of αβ-PFT pores aimed at deeper understanding of their properties and mechanisms of assembly. As shown in Supplementary Fig. 2, the mutant RL1 pore crystallized in this study translocates DNA very efficiently, thus providing a new paradigm for nanopore sensing applications.

## Methods

**Cloning and mutagenesis.** Genes encoding all lysenin mutants were generated by PCR mutagenesis[34]. All constructs were assembled in the pT7 expression vector[35] and verified by DNA sequencing of the entire gene insert. RL1 construct used in *Escherichia coli* expression (for crystallography trials) is an N-terminal fusion with DNA encoding STrEP (II) ligand followed by DNA encoding thioredoxin. A tobacco etch virus protease cleavage site is present in between the thioredoxin and lysenin. All RL1 mutants used in *in vitro* transcription-translation expression method do not contain any additional tags.

**Preparation of monomeric lysenin.** Lysenin mutant E84Q/E85K/E92Q/E97S/ D126G was used for crystal structure determination. Culture of *E. coli* BL21(DE3) pLysS cells harbouring the respective plasmid was grown in Terrific Broth, and protein expression was induced following addition of 0.2 mM isopropyl 1-thio-β-D-galactopyranoside and proceeded overnight at 18 °C. The cell lysate in 50 mM Tris/ HCl, pH 8.0, and 500 mM NaCl was loaded on Streptactin Sepharose resin (GE-Healthcare), and the protein was eluted with 50 mM Tris/HCl, pH 8.0, 500 mM NaCl and 1 mg ml$^{-1}$ Desthiobiotin (Sigma). The N-terminal tag was removed overnight at 4 °C in the presence of tobacco etch virus protease. Monomeric protein was further purified using gel filtration on Superdex 200 column (GE Healthcare) in 50 mM Tris/HCl, pH 8.0, and 500 mM NaCl, followed by the final step to remove the non-cleaved protein as well as cleaved tags by loading the sample on Strep HiTrap HP FT column (GE Healthcare) in the gel filtration buffer and collecting the unbound fractions. Lysenin monomer was stored at 4 °C.

**Preparation of lysenin pores.** Lysenin pore is made from monomer via a four-step purification method. First, monomer is oligomerized by incubating with liposomes containing SM and cholesterol (1:1, mol:mol) for 3 h at 37 °C. Resulting precipitate was collected by centrifugation and solubilized by addition of 10% DDM (w/v) in 50 mM Tris/HCl, 100 mM NaCl, pH 9.0, incubating for 3 h at 37 °C. After removing particulates by centrifugation, the supernatant was loaded on a Superdex 200 column equilibrated with 50 mM Tris/HCl, 100 mM NaCl, 1.0% DDM, pH 9.0. Fractions containing lysenin pore were identified by SDS–PAGE and the pore was further purified by loading on a Superdex 200 column equilibrated with 50 mM Tris/HCl, 100 mM NaCl, 0.01% DDM, pH 9.0.

**Crystallization.** Lysenin pores at a concentration 10 mg ml$^{-1}$ were crystallized at 20 °C using the vapour diffusion hanging drop method and from a solution containing 17% PEG-400, 0.2 M MgCl$_2$ and 0.1 M HEPES, pH 7.5. Before X-ray diffraction data collection at the synchrotron Elettra, crystals were cryoprotected in 30% PEG-400, 0.3 M MgCl$_2$, 0.1 M HEPES, pH 7.5, and stored in liquid nitrogen. To obtain heavy atom derivatives, lysenin pores were co-crystallized in the presence of 1 mM p-chloromercuribenzoic acid.

**X-ray data collection and structure determination.** Diffraction data were collected at XRD beamline at Elettra synchrotron, at 1 Å wavelength and 100 K. Data were processed to 3.1 Å in XDS[36]. There are two lysenin nonameric pores in the asymmetric unit of the crystal. Experimental phasing was performed with PHENIX (Autosol)[37]. 36 Hg sites were found, their geometry indicating that the lysenin pore is a nonamer, 18 sites per one pore (2 Hg atoms per protomer). Initial model was partially built by Autobuild (PHENIX). Further model building was performed using COOT[38] and refinement was done with REFMAC5 (ref. 39; Table 1). Electron density maps are shown in Supplementary Fig. 3. Ramachandran plot showed 93.8%, 5.4% and 0.8% of pore residues are in preferred, allowed and disallowed regions, respectively. Most of the residues in the disallowed region are in the flexible loops on the *Trans* side of the pore that are relatively poorly defined by the electron density. Structures were superimposed using COOT and figures were prepared using PyMOL[40]. The features of pores were calculated by Hole[41] and Hollow (channel inner surface)[42]. Electrostatic calculations were done using the Poisson-Boltzmann equation (APBS/PyMOL)[43].

**Non-denaturing mass spectrometry.** Lysenin was analysed by electrospray ionization mass spectrometry, introduced via gold-coated silica nanospray capillaries. Before analysis, lysenin was exchanged to a buffer composed of 200 mM ammonium acetate (pH 8) and 0.025% (w/v) DDM using a Micro Bio-Spin column (Bio-Rad). The spectrum was recorded on a Synapt G1 HDMS instrument modified with a linear drift tube and optimized for transition of large ions[44]. The capillary and cone voltages were 1.2 kV and 150 V, respectively. The protein was liberated from the DDM micelle by collisional-activation, using an accelerating voltage of 190 V and a gas pressure (Ar) of 0.2 MPa. Measured mass values are listed in Supplementary Fig. 6.

**HDX mass spectrometry.** HDX mass spectrometry was performed using Waters HDX manager composed of an automated sample preparation LEAP robot and a nano-Acquity UPLC coupled to a Synapt G2-Si mass spectrometer. The HDX reaction was initiated by a 13 × dilution into deuterated buffer containing 50 mM Tris, 100 mM NaCl adjusted to pH 9 using NaOD at room temperature for a time course of 20, 100 and 1,000 s for lysenin, whereas for the oligomer the deuteration buffer was supplemented with 0.025% DDM. Samples were sequentially quenched using 45 μl of hydrochloric acid (100 mM) and brought to pH 2.3. The protein was digested in-line using a pepsin immobilized column (Waters) at 20 °C. The generated peptides were trapped on a peptide trap for 3 min for desalting at flow rate of 200 μl min$^{-1}$ and then separated using a C18 column with a linear gradient 5–80% of acetonitrile and water both supplemented with 0.1% formic acid for 12 min at 0 °C at flow rate 50 μl min$^{-1}$. Sequence coverage and deuterium uptake were analysed using PLGS and DynamX programmes, respectively.

**AFM imaging.** AFM visualization of lysenin oligomers was performed by using Nanoexplorer (RIBM). AFM cantilevers with carbon nano-fibre probe (BL-AC10FS-A2) were purchased from Olympus. These cantilevers have a spring constant of 0.1 N m$^{-1}$ and a resonance frequency in the range between 500 and 600 kHz in water. Lysenin was obtained from Peptide Institute and diluted with a phosphate-buffered saline (10 mM, pH 7.5; Sigma) to a final concentration of 15 μM. The procedure followed in the preparation of the SM-containing lipid bilayer is explained elsewhere[30]. For high-resolution imaging, either 1.5 μl lysenin was preincubated with the SM/cholesterol (1:1) bilayer, formed on a 1.5 mm-diameter mica disk, or 10 μl lysenin was introduced into 70–80 μl Milli-Q water and the oligomerization of lysenin was followed *in situ* until the membrane was fully covered with oligomers. The excess lysenin was rinsed off and the AFM images of stable lysenin oligomers were recorded in Milli-Q water. The lysenin prepores and arcs were visualized in the incubation medium just after the assembly of oligomers into a stable hexagonal close-packed structure. AFM imaging was performed at room temperature.

**Molecular dynamics.** Molecular dynamics simulation and analysis of the data were performed with GROMACS 5.0.2 (ref. 45). Crystal structure of monomeric lysenin (PDB-ID 3ZXD) was solubilized in TIP3P water model[46] and CHARM27 force field[47] was used. Na$^+$ and Cl$^-$ atoms were added to a 150 mM final concentration before the simulation system was minimized and equilibrated for 0.1 ns at 310 K (V-rescale method[48]) and pressure of 1 bar (Parrinello-Rahman method[49]). Periodic boundary conditions were applied in all three dimensions. Long-range electrostatics were evaluated by the Particle Mesh Ewald method[50]. Simulation time was 1 μs. Protein structure image was prepared in PyMOL[40].

**Image analysis.** The asymmetric unit of the lysenin pore from two-dimensional electron crystallography, as previously reported[21] was rotationally averaged 9 × to reveal the same subunit arrangement as found in the pore crystal structure presented in this paper, using IMAGIC[51] software. The full symmetry was obscured by the P3 symmetry of the crystals but the resolution of subunit boundaries demonstrates that the correct symmetry has been applied. Application of no other symmetry resolves them, but smears out the features shown here.

**Double cysteine mutant experiments.** *Generation of proteins.* Two native Cys in the RL1 background were removed to generate RL1-AA. Monomers of RL1-AA and the double Cys mutant RL1-AA-V88C/Y131C were generated by coupled *in vitro* transcription and translation by using an E. coli T7-S30 extract system for circular DNA (Promega, no. L1130). As per the manufacturer's instructions, reaction mixture was supplemented with L-[35S]methionine (Perkin Elmer, no. NEG009A005MC) to radiolabel the protein.

*Disulphide bond formation.* Samples were spun in a microcentrifuge at 10,000 *g* for 10 min. and the supernatant was buffer exchanged into a reaction buffer (RB; 10 mM Tris, 50 mM NaCl, pH 8.0) using 0.5 ml Zeba spin columns (Fisher Scientific). Oxidation was achieved through incubation of the protein sample with Cu(II):1–10 Phenanthroline (500 μM:167 μM) for 3 h at 22 °C. Protein sample was buffer exchanged to RB.

*Haemolysis.* Sheep blood (Fisher Scientific, SR0051B) was centrifuged at 250 *g* in a microcentrifuge and the pellet was washed in an assay buffer (AB; 10 mM MOPS, 150 mM NaCl, 1 mg ml$^{-1}$ BSA, pH 7.4) repeatedly until the supernatant became clear. Pelleted red cell fraction was diluted to 1% (v/v) in AB. Haemolysis was carried out by mixing 10 μl of sample with AB up to 50 μl final volume and then adding 50 μl of 1% sheep red blood cell suspension on a 96-well clear bottom assay plate. Lysis of red blood cells was monitored at 37 °C with Bio-Rad Benchmark Plus plate (Bio-Rad) reader running Microplate Manager 5.2.1 software, collecting absorbance at 595 nm, mixing every 60 s. Upon completion of the programme, 5 μl of 100 mM dithiothreitol (DTT) was added to each well, and the programme was rerun.

*SDS–PAGE.* Lysenin monomers were produced through *in vitro* transcription and translation as described above and buffer exchanged into RB. Samples were then treated either with Cu(II):1–10 Phenanthroline (500 μM:167 μM) for 3 h at 22 °C, or incubated with 1 mM DTT for 3 h at 4 °C. A portion of each sample was then incubated with liposomes (composed of 0.275 mg ml$^{-1}$ SM, 0.325 mg ml$^{-1}$ phosphatidylserine, 0.55 mg ml$^{-1}$ phosphatidylethanolamine, 0.9 mg ml$^{-1}$ phosphatidylcholine, 0.45 mg ml$^{-1}$ cholesterol in 150 mM MOPS buffer pH 7, Encapsula NanoSciences), 20 μl liposomes per 100 μl of protein, for 1 h at 37 °C. A portion of oxidized, lipid-treated sample was treated with 5 mM DTT for 1 h at 22 °C. Samples were mixed 1:1 with Laemmli buffer containing 3% SDS, and heated to 70 °C for 15 min, before running on a 4–20% Tris HCl gel (Bio-Rad Criterion) in TGS running buffer. Gel was dried and subjected to autoradiography.

**Single channel current recording experiments.** Experiments were carried out using Oxford Nanopore Technologies MinION platforms. Recordings were conducted in 500 mM KCl at + 160 mV with a 5-s flick at − 160 mV every 1,200 s. 500 nM thrombin-binding aptamer was present in the buffer during the entire length of the experiment.

**Data availability.** Atomic coordinates of lysenin pore have been deposited in RCSB Protein Data Bank (PDB) under accession number 5EC5. The authors declare that all other relevant data supporting the findings of this study are available on request.

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

## Acknowledgements

M.P., N.R., M.K. and G.A. thank the Slovenian Research Agency for the financial support (Programme Grant number P1-0391). Part of the work conducted in the laboratory of G.A. was funded by the Oxford Nanopore Technologies. Part of this research was conducted by using high-performance computing resources at the National Institute of Chemistry, Slovenia (KI HPC). I.L. and C.V.R. acknowledge Dr Carla Schmidt and Dr Matteo Degiacomi for providing constructive scientific discussions, and Shane Chandler and Olga Tkachenko for HDX support. I.L., S.M., T M.A. and C.V.R. acknowledge financial support from the ERC IMPRESS (26,851) and the Medical Research Council (98,101). T.K. was supported by Integrated Lipidology Program of RIKEN and Grant-in Aid for Scientific Research 24,657,143 from the Ministry of Education, Culture, Sports, Science and Technology of Japan. We gratefully acknowledge financial support from the INSTRUCT (http://www.structuralbiology.eu/). The Oxford Division of Structural Biology is part of the Wellcome Trust Centre for Human Genetics, Wellcome Trust Core Award Grant Number 090532/Z/09/Z. The research leading to these results has received funding from the European Community's Seventh Framework Programme (FP7/2007–2013) under BioStruct-X (grant agreement N°283570). We thank the staff at the synchrotron facility Elettra in Trieste, Italy (beamline XRD), for support with data collection and fruitful discussions on crystallographic data analysis and the Centre of Excellence for Integrated Approaches in Chemistry and Biology of Proteins at the Jožef Stefan Institute (Slovenia) for initial robot supported crystallization screening.

## Author contributions

M.P., N.R., M.K. crystallized lysenin pores. M.P. determined the crystal structure. M.P., N.R., M.K., R.J.C.G. and G.A. analysed the data. M.K. performed molecular dynamics simulations. P.S. and R.H. purified proteins. N.W. carried out cysteine mutant experiments. N.W. and J.P. conducted single channel recording experiments. E.J.W. helped with molecular modelling. I.L., T.M.A. and S.M. performed mass spectrometry and I.L. performed detergent exchange protocols. C.V.R. supervised the mass spectrometry experiments. N.Y. collected and analysed AFM data with contribution of T.K. R.J.C.G. performed the electron microscopy analysis and constructed the phylogenetic tree. M.B., L.J. and G.A. conceived the project. M.P. and G.A. wrote the manuscript. All authors commented on the manuscript.

## Additional information

**Competing financial interests:** P.S., N.W., R.H., J.P., E.J.W., L.M., M.B. and L.J. are employed by Oxford Nanopore Technologies. Some of the subject matter discussed in this manuscript is disclosed in one or more patent applications filed by Oxford Nanopore Technologies. The remaining authors declare no competing financial interest.

