## [Peer Review File · Nature Communications]

Reviewer #1 (Remarks to the Author):

The authors have replied to most of the original comments, but there remain a few aspects of the manuscript that requires modification. The use of the lysenin pore as a DNA sensor, as has been highly developed for the *S. aureus* hemolysin pore, is the subject of the first paragraph of the results and the last paragraph of the discussion. Furthermore, 5 mutations were introduced into lysenin before its pore structure was solved, therefore it is not the native structure of the lysenin pore. These aspects of the manuscript makes it appear that, in contrast to the abstract, that the compelling reason to solve the structure was to develop a future DNA nanosensor, likely for DNA sequencing applications similar to alpha hemolysin, rather than to serve as a paradigm for an aerolysin family pore structure.

1. The state at the end of theintrosuction "This structure will be important for future applications of lysenin as a molecular tool in cell biology and, due to the unique shape of its pore, for nanosensing applications" could be made for many pore-forming toxins, such as the anthrax PA pore and the *S. aureus* hemolysin pores. Each pore will share fundamental features and exhibit unique features. In essence this statement could apply to any past and future pore structures.

2. Any reference to the nanopore usage should be eliminated (1st paragraph of the results and last paragraph of the discussion and any relevant figures). The focus should be maintained on the structure and what it tells us about the transition of this molecule to a pore complex, as the authors are suggesting this is the first pore structure of the aerolysin-like family of toxins.

We have removed the above-mentioned sections from the Introduction (last sentence of the introduction) and drastically shortened the Discussion section (third paragraph that was devoted to lysenin sensing applications). We now refer with the paper to lysenin relation to other aerolysin-like PFTs.

We, however, think that mentioning our attempts to develop lysenin as a DNA sensor are essential for understanding introduced mutations. Therefore, we propose to keep the sentences about DNA sensing in the Results section and propose to transfer panel a from Figure 1 to the Supplementary information (now Supplementary Fig. 2; as a result we have changed the numbering of panels in figure 1 and Supplementary figures). This result in addition provides functional data, in a sense, that show that the mutant pore is functional. In this way, it will be easier for readers to follow what was done and how we derived to the pore that was finally crystallized.

Reviewer #2 (Remarks to the Author):

The revised manuscript of Anderluh is considerably improved and with modest additional changes should be suitable for publication. In responding to the reviewers the authors clarified important data and rewrote key sections to change the emphasis appropriately. As such the results and introduction sections are appropriate with only modest trimming suggested for pages 10-11. The first 2 paragraphs of the discussion are satisfactory, the final one should be trimmed as the discussing of sensing applications is not quite relevant to the data at this time. Overall this represents an important advance, is now technically sound, and will be of interest to the readers of nature communications.

The mentioned section of the Discussion was trimmed as suggested. We have, however, left just one sentence indicating that lysenin pore may be used for sensing applications.

Reviewer #3 (Remarks to the Author):

The paper addresses an important question, that is, how do pore-forming toxins (PFT) assemble and form pores in target membranes? The study addresses this question with a high-quality crystal structure of lysenin, which is from the Aerolysin family of PFT. This structure is the first for this family. Alongside the crystal structure, the study also includes mass spectrometry (MS) data on the lysenin pore complex, including native MS and hydrogen deuterium exchange MS (HDXMS). Two-dimensional electron crystallography and atomic force microscopy are also used to establish the complex's stoichiometry. A small amount of electrophysiology data are also presented showing DNA translocation through the lysenin nanopore. This highly original work will be of broad interest to the nanopore and PFT fields. The paper is well written and easy to follow. My only suggestion is a relatively minor point.

It is revealed that the lysenin pore is one contiguous beta-barrel, which is stated as a possible advantage over other nanopores used in nanosensing applications, because there is no bottleneck or vestibule. My only concern is that looking at Figure 3a,b there still is a structural bottleneck in the pore-lining face of the channel. While not as pronounced as other PFTs, the narrowing of the barrel is still a noticeable feature of the pore. Also I am unsure why this lack of a vestibule is an advantage in the first place.

We agree with the reviewer and have changed the wording of the manuscript in the abstract (we have removed the statement that the lysenin pore represents a geometrically perfect tube; we have also significantly reduced the third paragraph of the discussion that was devoted to possible applications of lysenin pore for nanosensing applications that mentioned vestibule in the sense of advantages for sensing applications).

Referee #1 (Remarks to the Author):

The authors have expanded earlier work wherein they solved the crystal structure of the soluble monomer of lysenin and now have solved the structure of the lysenin pore complex with a derivative containing 5 mutations within the β -barrel complex. Lysenin is likely part of the ancient immune system of the earthworm, something that has been described by others. It is apparent from the pore structure that lysenin is a member of the small β -barrel pore forming toxins such as aerolysin and *Staphylococcus aureus* α -hemolysin. Although its structure is suggested to be aerolysin-like by a DALI analysis its mechanism seems to be more similar to that *S. aureus* α -hemolysin. Lysenin forms a β -barrel pore that is composed of 9 monomeric units and its pore size is consistent with that found for other small β -barrel pore formers, which are often heptamers. The major structural change in the monomer shown herein, when it undergoes the transition to the pore, is the unfolding of one β -strand and the refolding of an irregular loop structure, which together form the single amphipathic hairpin of each monomer that combine to form the β -barrel pore. From the way the manuscript was written it is not entirely clear if the focus was to understand the structure of the lysenin pore or to develop it as a DNA-translocating pore, the latter of which only comprises a small part of the manuscript. Generally the work shows invertebrates, or at least the earthworm, have evolved or acquired a protein that utilizes the same fundamental approach employed by many other pore-forming toxins to assemble a β -barrel pore.

1. Since so little was developed with respect to the DNA translocating aspect of this pore why solve the structure of the penta-mutant rather than native lysenin? Did the native protein not crystallize? It is possible that the negative charges within the β -barrel pore could have altered its shape somewhat due to charge effects.
2. The authors indicate that this is another member of the aerolysin family of small pore forming toxins. This is difficult to imagine since (1) it exhibits no structural similarity with aerolysin (other than it is mostly β -sheet) or any of its closer members, (2) there is little primary structural similarity, and (3) it does not require proteolytic activation wherein a propeptide is released. There is ample evidence that the formation of a membrane spanning β -barrel by the oligomerization of monomers with 1 or 2 amphipathic hairpins is a common mechanism among many unrelated pore-forming toxins from all kingdoms of life. Therefore, the structure of lysenin extends this concept to invertebrates, but the authors dialog comparing lysenin to aerolysin or its pore forming mechanism (as described in ref 19) is not relevant except in the broadest of terms. If anything, its mechanism (even though the DALI analysis points at the aerolysin family) resembles that of *S. aureus* α -hemolysin more than that of aerolysin.
3. The beginning of the data section is odd, as the authors begin with the DNA translocation experiment in extended figure 2. Logically, they could not do this experiment, wherein they engineered specific mutations within the β -barrel pore, without first knowing the pore structure. Yet they solved the structure of the lysenin with 5 mutations within the β -barrel. How did they authors select these mutants without first knowing the structure or the location of the membrane spanning hairpin? Was there a prior study that mapped this region or did they first get a structure of the native protein?
4. In Figure 3e, the two middle panels: it is not clear if the authors reduced the disulfide before or after the monomers had assembled into a prepore complex. If the latter, it would be more convincing to follow the change in lysis with the prepore cysteines oxidized before injecting reducing agent to unlock the prepore and continuing to follow lysis.
5. In the abstract the authors indicated that understanding the mechanism of lysenin, a toxin from the earthworm, will be important "...to future applications in the prevention of disease...". This statement makes no sense, as this toxin is not involved in human disease, and there are pore structures and structure-function data on many bacterial toxins that are much more relevant to disease.
6. The authors also suggest in the abstract that this structure will make it "prime candidate" for future

nanosensing developments. There is little data to support the belief that this pore forming toxin, among many others whose structure are already known, will make any it any better in these putative applications.

7. The overlaid structures in Extended figure 1a are too small to see any significant similarities.

8. The data in Figure 1a is more appropriately described in the Methods section and should be eliminated from the figure. The AFM in 1c is not a height AFM since no height data were included with the image. Figure 1b and c are superfluous in view of the atomic resolution pore complex in 1d.

9. In extended Figure 2 the authors engineer the pore to decrease the negative charge, which appears to quiet the fluctuations in the pore conductance. They add thrombin binding aptamer (TBA) to a single channel and suggest the downward spikes in conductance (pore blockage) reflects translocation of the aptamer through the pore. No controls are included to show pore conductance in the absence of the TBA or the effect of adding thrombin, which binds the aptamer and should lengthen the time the pore is closed. Many more studies need to be done to show this translocation capacity and its features: it would be more appropriate to expand these data into a distinct manuscript that focuses on this aspect of the pore.

10. In reference to the above comment, the noisy wildtype pore is unlike that observed by Ide et al (BBRC. 346: 288-292). Ide et al. showed the lysenin channels are stable and no noisier than the mutant lysenin shown herein. It appears that there is a significant difference in the voltage applied across the membranes in the two studies and that if the authors lowered their voltage to 40-60 mV, similar to Ide et al., this noise would be eliminated. Ide et al. observed increased noise as the voltage was increased to 80 mV, which suggests that the noisy channels of the WT lysenin pores observed by the authors herein are due to the 160 mV applied across the membranes.

11. The authors indicate that there is a dramatic reorganization of the pore forming domain upon the transition of the prepore to the pore: in reality the bulk of the pore forming domain remains intact and only the residues associated with the hairpin undergo reorganization. Similar observations have been made for the *S. aureus* α -hemolysin. Furthermore, the 45^o change in the relationship between the N-and C-terminal lobes was previously shown (by some of these same authors) to be largely due to binding the sphingomyelin receptor in a co-crystal of the soluble lysenin monomer bound to its receptor sphingomyelin.

12. The authors state in the introduction that "Although some pores are available with monomeric crystal structures, the lack of pore structures with atomic resolution has limited the use of $\alpha\beta$ -PFTs in such applications" (i.e., nanobiotechnology and medicine) and then go on to show that 6 other structures are known in Extended Figure 4. It seems that there are plenty of nanopore structures that exhibit pores of a similar size to that of the lysenin: there does not appear to be a dearth of these structures, as suggested, so the addition of the lysenin pore to the repertoire does not greatly enhance the already available nanopore structures. Furthermore, there are also a number of atomic level structures of porins and oligomeric α -helical pore forming toxins available. Thus, there are many pore structures available that are being or could be engineered and implemented in such studies.

Referee #2 (Remarks to the Author):

The manuscript of Pobobnik and co-workers provides a novel structure of the lysenin pore of an alpha/beta pore forming toxin used as a bacterial defense mechanisms in earthworms. Such pore structures not only have medical interest due to their role as anti-bacterial virulence factors, such pore structures have recently gained attention and interest as DNA sensors in nano-biotechnology applications. The data of this paper adds important insights in the context of both fields.

The structural and functional data on the lysenin nonamer pore and the details of its assembly from monomers nicely clearly lays out a mechanism of sphingomyelin (SM) dependent assembly mediated by the c-terminal domain of lysenin while clearly illustrating the significant conformational rearrangements of the N-terminal domain required for pore formation. The disulfide-containing mutant abolishes pore formation as expected and mass spectrometry and AFM measurements provide ancillary support for the pore structure in solution. Although many structures of pore monomers are available, defined pore structure are still few and this study providing a plausible structural mechanism for pore assembly with quite extensive structural characterization represent a welcome contribution that will be of interest to a very wide audience.

The overall picture is quite positive, however some data is less compelling and requires significant explanation or revision before the manuscript and its conclusions can be further evaluated. In addition the organization of the manuscript could be improved.

Specific Comments

The paper has a very well structured and clear introduction, and introduces the concept of the high stability of the pore (e.g. stable at high temperature in SDS). However, the manuscript then veers into a discussion of the pore's potential role as a DNA sensor, the failure of the WT pore to function as such, and the generation of a mutant pore structure (with removal of Asp/Glu residues) that does function as a sensor, and is the basis of the paper's main structural studies. This sequence in the discussion detracts from the main points of the paper. The authors sincerely wish to make contributions to the DNA sensor field but the paper has only modest contributions along these lines. It is assumed that the authors attempted the engineering of the construct in order to provide a species suitable for crystallography, the authors have decided to not discuss these aspects of the project development. However, it would be important for the field for these details to be laid out in the methods etc. Thus, some data and discussion along these lines is suggested.

Heterogeneity of the construct using MS.

Native MS is used to understand the protein oligomerization state. A nonamer is seen and dissociation of the nonamer into octamer and monomer species is shown. These data are relatively convincing. Missing from the methods are details of the instrument resolution and accuracy for these studies, the discrepancy (if any) between the construct mass and the observed mass. The notable absence of any adducts is of interest, are there any buried in the grass etc.?

Crystallography

The xtal structure data appears quite satisfactory in terms of model quality and resolution. The nonamer structure is well supported by MS and AFM data. The structural discussion is quite coherent and lays out how the distribution of charged and hydrophobic residues is consistent with the packing into the membrane and specific interactions with SM.

HDX

The HDX data is quite unsuitable at present. The significantly varying coverage makes appropriate comparisons between monomer and nonamer quite problematic. No progress curves for HDX that clearly define the observed peptides are presented which would help to better define differences when peptide are found for both monomer and nonamer. No detailed comparison to the structure (e.g. # protons exchanged) are made in the cases where coverage is present that would indicate relative stability of helices etc. In some cases single residue data is potential indicated (F70?) but how that is deconvolved from overlapping peptide and their exchange is not defined.

On the other hand, the authors do have quite potential mapping (although they don't really point it

out) of the overall domain architecture based on the varying proteolysis results (e.g. coverage maps). Limited proteolysis would be much more compelling in this section as that would provide clear delineation of the regions of extra stable beta-sheet segments that form the pore vs. the likely digestible C-terminal segment.

The structure section after the HDX section is quite clear and convincing and begins to lay out their view of the assembly process. The authors should be cautioned that this is a proposed assembly mechanism, although they provide good evidence for the structure per se, the steps and details of the assembly are not yet confirmed.

Lines 192-194-appears to confuse kinetics and thermodynamics, e.g. there are kinetic barriers to assembly which are different from the overall stability features of the pore.

Hemolysis and SDS Gel

Data on the double mutant is important to the story. This links the structural state to the proposed functional switching that must occur for the monomer-nonamer transition. The behavior of the oxidized and reduced forms of the double mutant in the hemolysis assay is appropriate and reasonably compelling. However, the SDS gel results are less conclusive. For example the gel nicely shows the SM dependent conversion of RL1-AA and the double Cys mutant from monomer to nonamer and illustrates the very homogenous nonamer species that provides source material for crystallization trails. Other parts of the gel or the labeling and/or the figure legend and text descriptions are confusing. Lanes are labelled +/- DTT while the gel header also used reduced and oxidized labels. This creates confusion re; order of addition and final state definitions. +DTT should reduce oxidized species and permit oligomerization, is the argument there is less monomer in -vs + DTT lanes for the mutant? This likely needs to be reworked or redone.

Figures:

1c needs explanation of AFM color coding

2e relative fractional uptake a poorly defined and weak structural measure

3b perhaps a side by side vs super-imposed would show the conclusion more clearly?

3e-see comments above

Referee #3 (Remarks to the Author):

The manuscript by Podobnik et al. entitled "Crystal structure of an invertebrate cytolysin pore reveals a unique set of properties and mechanism of assembly" deals with a new crystal structure of the invertebrate pore forming toxin, called Lysenin. This is the first structure of a pore forming toxin in the aerolysin family of such toxins. A variety of other biophysical techniques (including mass spectrometry and atomic force microscopy) are used to establish the 9-mer stoichiometry observed in their crystal structure. Further biochemical assays (including red cell lysis and SDS PAGE) are used to establish the mechanism of pore formation. The crystal structure is high quality and serves to provide insight on the structure of aerolysin-type pore forming beta-barrel toxins. The manuscript is generally well written and easy to follow.

Below I highlighted some concerns about the manuscript.

(1) The authors convincingly demonstrate that the oligomer is a nonamer, although the 9mer was

produced in vitro and not extracted from cell surfaces. So there is some question regarding whether the oligomer is octameric on cell surfaces, but is nonameric when assembled in vitro.

(2) For the assembly mechanism, the authors use AFM and witness arcs of partially assembled monomers. These may either be assembly intermediates or disassembly intermediates. To establish assembly, kinetic time points would need to be taken showing these arcs form before fully assembled lysenins are observed.

Minor points

Hemolysis assay figure legend (Fig. 3e) and figure are impossible to follow as written and drawn.

Redefine "TBA" abbreviation in Fig. 2 legend of Extended Data.

Line 373. Insert "was" after "pore".

Line 377. Replace "was concentration" with "at a concentration of"